# Forest Bioeconomy in Brazil: Potential Innovative Products from the Forest Sector

Yasmin Imparato Maximo [1,*], Mariana Hassegawa [1], Pieter Johannes Verkerk [1] and André Luiz Missio [2,3]

[1] European Forest Institute, Yliopistokatu 6B, 80100 Joensuu, Finland
[2] Programa de Pós-Graduação em Ciência e Engenharia de Materiais (PPGCEM), Centro de Desenvolvimento Tecnológico, Universidade Federal de Pelotas, Rua Gomes Carneiro, 1, Pelotas 96010-610, Rio Grande do Sul, Brazil
[3] Programa de Pós-Graduação em Ciência Ambientais (PPGCAmb), Centro de Engenharias, Universidade Federal de Pelotas, Benjamin Constant, 989, Pelotas 96010-020, Rio Grande do Sul, Brazil
* Correspondence: yasmin.maximo@efi.int

**Abstract:** The forest sector plays an important role in the circular bioeconomy due to its focus on renewable materials that can substitute fossil or greenhouse gas emissions-intensive materials, store carbon in bio-based products and provide ecosystem services. This study investigates the state of the bioeconomy in Brazil and its forest industry. Specifically, this study presents some examples of novel wood-based products being developed or manufactured in Brazil and discusses possible opportunities for the development of the country's forest sector. The pulp and paper industry plays an important role in the forest sector. It has also been showing advancements in the development of cascading uses of wood invalue-added products, such as nanocrystalline cellulose, wood-based textile fibers, lignin-based products, and chemical derivatives from tall oil. Product and business diversification through the integration of the pulp and paper industry to biorefineries could provide new opportunities. Moreover, biochemicals derived from non-wood forest products, such as resin and tannins could promote diversification and competitiveness of the Brazilian forest industry. Although some engineered wood products are still a novelty in Brazil, the market for such products will likely expand in the future following the global trends in wood construction.

**Keywords:** Bioeconomy; wood-based products; forest-based sector

## 1. Introduction

The development of the circular and sustainable bioeconomy is considered a means to address many challenges of modern society, such as the dependence on fossil fuels and non-renewable materials, food production, and environmental degradation [1–5]. Bioeconomy refers to the use of sustainable biological resources and the application of biotechnological knowledge that, through different processes, are able to provide products and services [6,7]. The bioeconomy has been linked to the concept of circular economy, which focuses on using waste materials as resources and favor a high degree of recycling, thereby maintaining resources and materials in use for as long as possible [8–11].

Bioeconomy is a broad and complex concept whose development is affected by several forces in the social, economic, and environmental spheres. Some of the drivers that shape the bioeconomy are the following: resources availability, research and development in biotechnologies, demographics, policies, and regulations [12]. Drivers of the bioeconomy include factors associated with supply (e.g., technology and innovation, market organization, changes in biomass supply), resource availability, demand (e.g., demographics, economic development, and consumer preferences), and measures by governments to influence the development of the bioeconomy (e.g., policies, strategies, and legislation) [9]. Investments in research, intellectual property rights, and public

acceptance are also considered important drivers for the development of innovations within the bioeconomy [6].

The number of strategies and initiatives to develop the bioeconomy have been increasing worldwide [13]. These actions were first observedin developed countries, but, recently, developing countries have also been setting up bioeconomy strategies and actions [14]. In general, bioeconomy strategies typically take into consideration the country's reality, challenges, and interests, which could be food security, protection of biodiversity, development of biotechnology, climate change mitigation, among others [15,16].

Forests and the forest sector are important parts of the land-based bioeconomy [17–19]. Forests provide raw material for numerous types of products and are also important for climate change mitigation through their capacity to remove carbon dioxide from the atmosphere and store carbon in forest biomass and soils. Besides raw material provisioning and carbon storage, other important ecosystem services are provided by forests, such as air purification, habitat provisioning, biodiversity protection, as well as watershed and soil protection [20]. Wood is an important raw material from forests that is used for producing sawn wood, panels, and boards, as well as pulp and paper for a wide range of applications. Besides these traditional products, the forest sector has been focusing on increasing the use of residues and industrial side streams as raw material and increasing its product portfolio with innovative value-added products that have lower negative impacts to the environment [21–24]. Forest-based raw materials can be extracted and transformed into many products that can be used to substitute products made from fossil or resource-intensive materials [25], with notable developments in the field of biochemicals, cellulosic fibers, biofuels, and the construction of buildings [23,24].

Countries with a developed forest sector are in a favorable situation, having the opportunity to further develop their economies while taking measures that contribute to climate change mitigation by storing carbon in wood products. Brazil is one such country and, because of its vast land area and its advantageous geographical position, it has the potential to become a key actor in the global forest bioeconomy context [26,27]. The country is rich in biodiversity, with numerous wood and non-wood forest products from native forests that could enhance the country's transition to a circular bioeconomy when sustainable forest management is combined with value-added processing and innovation [28–31]. In addition, the forest sector is constantly investing in technology and management of planted forests. In 2019, the sector invested around USD 12.5 million (equivalent to around BRL 50 million, according to the average exchange rate of 2019 [32]) in innovations in the sector, half of which was spent on research and development [33].

Brazil has the second largest forest area in the world, covering 496 million hectares, with over 485 million hectares of naturally regenerating forests and around 10.5 million hectares of planted forests [34,35]. Although the majority of the country's forest area is natural, around 78% of the wood used by the industry comes from planted forests owned by the forest industry [35]. The planted forest areas are mainly composed of exotic species *Eucalyptus* spp. plantations, which represent the largest share (about 76%) which are mainly located in the southwest and midwest, while *Pinus* spp. plantations represent nearly 20%, mainly located in the southern region [33,36]. Most of the pulp and paper companies in Brazil are located in the southwest and southern regions, where mostly eucalypt and pine are used as raw materials. Moreover, the majority of the panel industries are located in the southern region, where they consume mostly pine wood [33,35].

The wood products value chain of the companies associated with the Brazilian tree industry (IBÁ) contributed 1% of the national gross domestic product and created 1.5 million direct jobs [37]. The main products from the Brazilian forest industry (in terms of relative share of generated taxes) are pulp and paper (61%), wood panels and engineered wood flooring (25%), solid wood products (8%), and others (6%) [36].

Brazil's planted tree industry is the second largest producer of wood pulp in the world. The sector produced an average of 20 million tons per year between 2017 and 2020, around 75% of which were exported [34,37]. Wood pulp and other traditional forest

products (e.g., sawnwood and wood chips) are internationally traded commodities and are highly dependent on international supply and demand. As their competition is essentially based on price, this market could represent a risk to the suppliers [38,39].

The current bioeconomy in Brazil involves all economic activities stemming from innovation in the field of biological sciences and culminating in the development of more sustainable products, processes, and services through biotechnology. Besides the added economic value from the implementation of a bioeconomy strategy, the bioeconomy in Brazil is also considered to contribute to complying to international agreements through these actions [40], such as the United Nations' 2030 agenda [41] and the Paris Agreement [42]. If Brazil maintains its commitment to recover degraded lands and to reduce greenhouse gas (GHG) emissions [43], the country will be able to further strengthen the forest sector while addressing these environmental issues and moving towards a circular bioeconomy.

Besides being widely discussed within government and politics, bioeconomy has also been increasingly generating interest as a research topic [18,44]. The number of published scientific studieson innovations in forest bioeconomy increased between 2010 and 2021. However, most of these publications focus on Europe and North America, with around 80% of the publications, while South America has a small representativeness of about 6% on the topic [45]. Thus, it is important to synthesize the available information on the forest bioeconomy for underrepresented countries, such as Brazil, and global regions. Although Brazil manufactures several products within the traditional forest sector, such as wood pulp, fuelwood, and wood panels, there is limited and scattered information covering the recent product developments connected to the forest bioeconomy in the country. Moreover, the country's forest sector still struggles to reach segments with higher levels of industrial processing [46]. This can be related to the fact that Brazil, as other developing countries, adopted actions such as reducing costs, improving production techniques, scale economy, tax incentives, and others focused on commodities such as cellulose and sawn wood, as a means to increase its competitiveness [47].

Therefore, in this study, we aimed to understand the current state of the forest bioeconomy in Brazil, focusing on the developments in planted forests, and to review new wood-based products developed or manufactured within the Brazilian forest bioeconomy with market potential.

## 2. Methodology

To understand the role and prospects of the forest sector in the current Brazilian bioeconomy, desk research of official documents (e.g., policies, action plans) and activities and innovative products was conducted in 2021. Because around 78% of the commercial wood comes from planted forests [35], this review focused on documents dealing and innovative products coming from the planted tree industry. The official documents that were reviewed had to be in force at the time and explicitly focus its subject to bioeconomy. Regarding the review of the products, as the objective of this paper was to present new forest products associated with the bioeconomy, traditional products such as sawn wood, paper products and fuelwood were not considered in the review. Instead, we focused on both technologies and innovative products that are either entering the market or that have the potential to increase in market share in the near future.

The selection of the new products for the analysis was performed by a compilation of a list of products and technologies under development or production within the Brazilian forest sector. These products should differ significantly in their characteristics from the traditional products manufactured by the sector. This list was based on two complementary approaches, i.e., literature search using online search engines (e.g., Google, Google Scholar) and direct search on websites of major manufacturing companies. As products under development are often not widely advertised or associated with scientific papers, an exhaustive search and manual filtering of the findings was necessary for a successful compilation of the developments. Therefore, the following approach was used to

identify recent technologies and product developments that could provide insights regarding the market development in Brazil. A set of web searches using several combinations of blocks of words, both in Portuguese and English, were performed as follows:

(a) Keywords block one–place; Brazil OR Brazilian
(b) Keywords block two–object; Forest products OR forest developments OR bioeconomy OR bioproducts OR forest industry
(c) Keywords block three–adjective; new OR innovative OR recent

In addition, a targeted search was performed using well-known product names, product categories, or sectors. An example of a search targeting one product type is as follows:

(a) Keywords block one–place; Brazil OR Brazilian
(b) Keywords block two–object; cross-laminated timber OR CLT OR engineered wood products

In addition to the outlined search, a list of the most important companies active in the Brazilian forest sector was compiled. The list of 47 member companies of IBÁ, the association representing private and public organizations working with planted forests, was used for this purpose.

Having the full list of potential products compiled after the web search, the products to be reviewed were selected based on two criteria. First, the product had to be produced by the Brazilian planted forest industry, and second, preference was given to the products considered market attractive. Market attractiveness was based on (i) whether the product could enter the most promising global markets for emerging wood products, such as biochemicals, textiles, construction, biofuels, packaging, and bioplastics [23]; (ii) the product technology readiness level (TRL) [48]. TRL provides a consistent method of assessments and comparison of maturity of different types of technology [49]. In this study, focus was given to products with TRL ≥ 6, which indicated that products in development had reached a prototype demonstration level in a relevant environment, as products beyond this level are estimated to enter the market in a relatively short time [22]. As the TRL was not disclosed by the organizations for all the selected products, it was estimated based on publicly available information and the description of each level provided by the European Commission [48].

The products that met all the criteria were grouped by their industry type and respective value chains: pulp and paper industry, solid wood products and panels industry, and biorefineries. After grouping, the products were distributed in a biomass value pyramid to obtain a better understanding of the developments of innovative wood-based products in Brazil regardless of the industrial sector. The selected products were distributed in the pyramid according to their type in an adapted version of the forest bioeconomy value pyramid proposed by [50].

## 3. Results

### 3.1. The Brazillian Bioeconomy

Until recently, Brazil had some policies fostering the bioeconomy but no integrated strategies focused on the subject [51]. In 2016, the National Strategy for Science, Technology and Innovation was created. The National Strategy, in force until 2022, is a medium-term guide that aims to assist in the preparation, conduction and monitoring of actions in science, technology and innovation in the bioeconomy [52]. In 2018, the Action Plan for Technology and Innovation on Bioeconomy was established to implement the National Strategy for Science, Technology and Innovation and to ensure the promotion of a national bioeconomy by creating governmental bodies specifically for this area [40]. The Action Plan aimed to promote social, economic, and environmental benefits, filling essential knowledge gaps, fostering innovation and providing conditions for the strategic insertion of the Brazilian bioeconomy within the global scenario [40]. The Action Plan was divided into three main action fronts: 1)the scientific and technological development for the

sustainable production of biomass, including the use of residues, as well as the genetic improvement of native species for bioproducts; 2) furthering of innovation in bioindustries through the scientific and technological development for biomass processing; 3) the development and manufacture of high added value bioproducts, especially chemicals from biomass, and aimings to consolidate the circular bioeconomy.

### 3.2. New Forest Products in Brazil

Worldwide, there are several technologies and processes that could improve the value of forest products within the forest sector [45]. Many industries have increased their competitiveness and environmental performance due to developments and innovations of value-added forest products [53]. Companies are investing in the development of new bio-based products that represent more sustainable alternatives to traditional products [33]. The investments in innovation aim to add value to forest products and better use industrial side streams by developing the value chains, and producing high-value-added bio-based products, such as biofuels, bio-oils, nanofibers and wood-based textiles [33]. Yet, transitioning from low-value-added to high-value-added products can be challenging in a forest bioeconomy [50]. Figure 1 presents the product categories and the innovative forest products in Brazil that were included in our review, according to the biomass value pyramid [50].

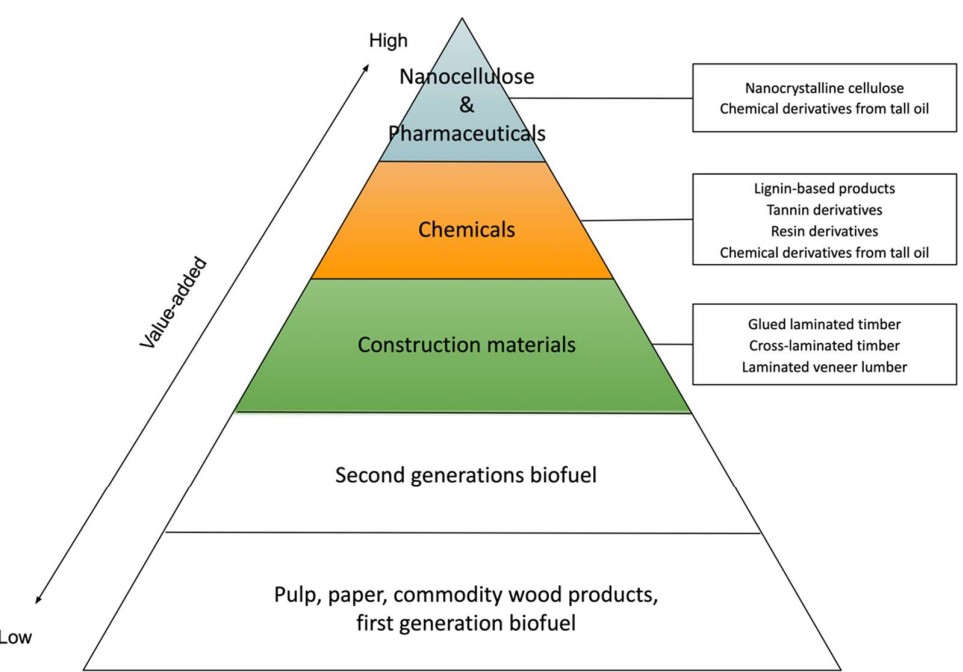

**Figure 1.** Forest biomass value pyramid (adapted from [50]), and innovative forest products manufactured in Brazil (in boxes).

Currently, the main innovations in forest products in Brazil are at the top three levels of the pyramid, with the two bottom levels either not having any recent developments or focusing on improvements of existing traditional products. The products positioned at the top three levels of the pyramid, shown in the boxes, are further explored in the following sections, grouped according to the main forest industries: pulp and paper, solid wood products and wood panels, and biorefineries.

### 3.2.1. Pulp and Paper Industry

Brazil has been investing in the development and manufacture of innovative products (Figure 2). Some of these products (e.g., eucalypt kraft fluff pulp, dissolving pulp) have demonstrated commercial importance worldwide, from food to pharmaceuticals. The following products have potential to enter the market in the coming years and will be discussed in the next sections: nanocrystalline cellulose (NCC), wood-based textile fibers, lignin-based products, and chemical derivatives from tall oil.

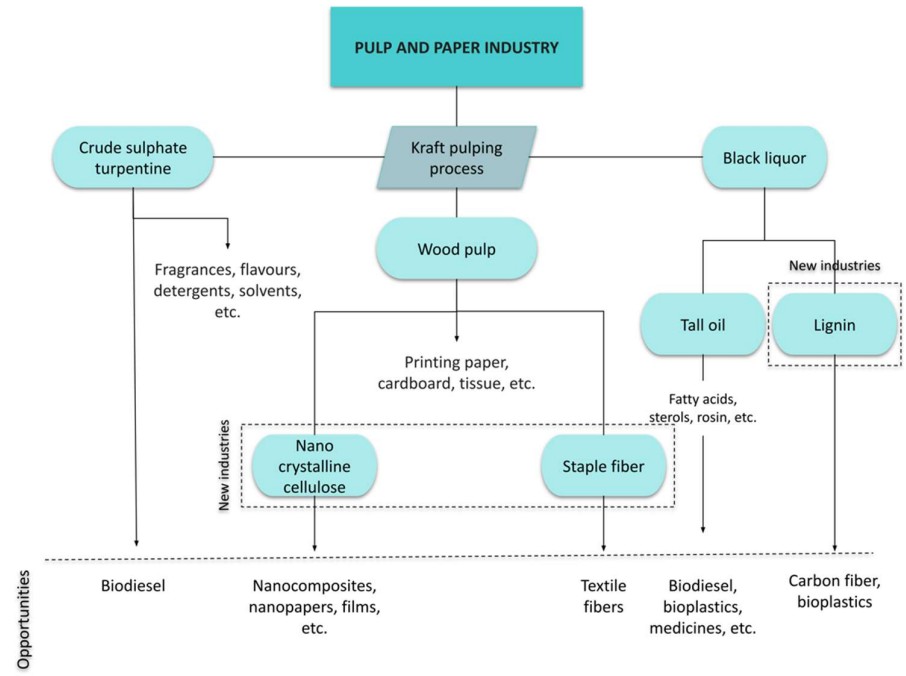

**Figure 2.** Main innovative products from the Brazilian planted tree pulp and paper industry, and potential new industries.

### Nanocrystalline Cellulose

NCC has been attracting interest due to its physicochemical properties [54], it being suitable for application to several segments, and it serving as a sustainable material [55]. One possible application is in the packaging industry as a reinforcing material for polymers in nanocomposites, producing nanopaper and barriers films, among others [55,56]. Besides improving strength, brightness, barrier property, and weight, among others, the application of NCC can have an antimicrobial capacity, which is a useful property for food packaging [56]. Nanocellulose is also being applied to the development of biomaterials to substitute traditional fossil-based polymers [57]. Further applications are seen in the pharmaceutical, food, textile industries, as well as some other industries [55]. Due to the versatility of nanocellulose, its use has been increasing. In 2019, it was estimated that its global market would grow in volume at a 30% annual rate until 2025 [58].

Even though NCC has been studied in Brazil since the early 1990s, the topic has only recently attracted greater scientific and industrial interest [59]. Cellulosic pulp from eucalypt is largely researched as a source for NCC in Brazil [60–64] for which different production techniques have been developed. Brazilian research studies target applications of NCC for the improvement of paper properties [65], development of biofilms for packaging [66], and for pharmaceutical applications, such as medicine beads [67], wound dressings [68,69] and hygiene products [70]. There are two pilot-scale nanocellulose plants in Brazil [70]. One of these pilot plants supplied cellulose microfibrils for hand sanitizers during the COVID-19 pandemic [71].

Considering the status of cellulosic pulp production in Brazil, the production of NCC could be a good opportunity to develop one of the main segments in the forest sector, as this material can be an important component in the paper and packaging industry [56]. NCC could also be a useful material for the pharmaceutical industry, where it has the potential to improve characteristics of products and replace the use of fossil-based products [72]. For instance, NCC could substitute fossil-based carbon nanotubes, which are used as a vehicle for drug delivery [73]. Moreover, NCC-based hydrogels can be an alternative for the traditional silica-based material in the production of drug carriers, where the alternative shows an improvement in biodegradability and biocompatibility [73]. Microspheres composed with NCC also present advantages such as encapsulation efficiency, and sustained drug release [74]. Also, significant cost reduction is observed when manufacturing NCC-based wound dressings from wood sources, when compared to the bacterial NCC product [68]. Thus, as NCC presents several current uses, and future applications are now under study, this segment could become an important opportunity for the increase in value-added products from the pulp industry.

As with the production of most chemicals, the environmental impacts of nanocellulose production depend on the chemical modification and the mechanical treatment route used in the manufacturing process [75]. However, the major environmental impacts associated with nanocellulose production are due to the high demand for process energy [76]. For instance, one energy-intensive step is the release of nanocellulose from native cellulose fibers by breaking the hydrogen bonds [75]. Regarding the land use impact, it may be associated with the source of chemicals used in the process. For instance, ethanol is used as a solvent and is responsible for an important environmental impact. According to a lifecycle assessment (LCA), ethanol from corn has a larger environmental impact on land than ethanol from ethylene from fossil sources [75]. When it comes to the impacts at the end-of-life of products containing nanocellulose, according to studies, no environmental risks are associated with its presence in surface water [77], nor with biodegradation and ecotoxicity [78,79].

Wood-Based Fibers for Textiles

Currently, the main raw material used in the production of textile fibers is petroleum, to manufacture fibers such as polyester, acrylic, and nylon. However, new technologies for producing wood-based textiles have been gaining market interest in the past few years [80]. These innovative materials can be used as a substitute for certain fossil-based counterparts or even other types of unsustainably sourced plant-based fibers, such as cotton [81,82]. Considering that the textile sector is one of the largest industries in the world, wood-based textiles could have an important role in climate change mitigation [25,83].

Wood-based textiles are usually produced from man-made cellulosic fibers, such as viscose and lyocell [81]. The production process of viscose uses traditional technologies and is based on dissolving wood pulp and wet spinning [81,84]. In 2019, 585 thousand tons of dissolving pulp were produced in Brazil, most of which were exported to China, the United States, and Europe [85,86]. Currently, Brazil has two dissolving pulp mills that produce fibers that can be used for viscose, lyocell and other textiles. In 2018, a joint venture of the Brazilian company Duratex and the Austrian company Lenzing was launched with plans to build a dissolving pulp factory in southeastern Brazil with an annual production capacity of 450 thousand tons of dissolving pulp [87,88]. The pulp to be produced in Brazil, using eucalypt wood, will be exported to Asia, where it will be used to manufacture lyocell. Besides the Duratex Lenzing dissolving pulp mill in planning, Bracell also announced investments. The company already has two dissolving pulp plants operating in Brazil in the northeast and southeast regions. In 2019 the expansion of the processing plant located in the southeast region was confirmed, where one of the largest dissolving pulp mills in the world will take place, with an estimated production capacity of 1.5 million tons of dissolving pulp per year [89].

There are other types of wood-based fibers for textiles in several stages of development and production worldwide. These new wood-based textile fibers use innovative production processes that are considered better for the environment, using non-toxic chemicals for dissolving pulp [90]. Even though these new wood-based textile fibers are not yet produced in Brazil, there is the possibility, even if in the medium to long term. The Brazilian company Suzano Papel e Celulose, one of the largest wood pulp producers in the world, is a shareholder and the main supplier of wood pulp to Spinnova, a Finnish company that has developed an innovative and sustainable process to manufacture wood-based fibers for textiles [91].

The environmental impacts of wood-based textile fibers depend on the source of the raw material, the fiber type, and the production process [92]. Lyocell, for example, in general has lower environmental impacts compared to cotton, synthetic fibers and even other wood-based fibers of older technology, such as viscose. The main reasons are the use of renewable energy to manufacture the fibers, the use of less chemicals and water, and the lower GHG emissions [82]. When it comes to the impact on land use, lyocell also has a smaller impact than cotton as it requires less land area to produce the raw material [82]. Wood-based textile fibers produced in Brazil would also have the advantage of reducing the pressure on land use due to the higher biomass yields compared to fibers produced in temperate and boreal regions.

Lignin-Based Products

Another product segment that is currently in the developing stages relates to lignin-based products. Lignin is the second most common component in biomass, after cellulose, and is an abundant by-product of the pulping process, but still with limited use [93]. From the 70 million tons of lignin produced worldwide every year, only around 5% are used in commercial applications, the rest being used for energy through direct burning in the industry [93]. Currently, there are several applications and products worldwide that use lignin as feedstock, such as synthesis gas biofuel, binders, adhesives, coatings, and dispersants [94,95], which are traditionally fossil-based products. Besides that, further products with innovative uses and applications of lignin are in development, such as foams [96], carbon fibers [97], and rechargeable batteries [98].

Derived as a by-product from the pulping process, lignin extraction and processing could become a trend in Brazil. The first industrial-scale plant was launched in 2019 with the capacity to produce 20 thousand tons of lignin per year. The lignin industry is fed by an industrial side stream of the pulp processing of eucalypt wood, which is the main production line of the company [99]. Lignin can be used in several industries and different lines of products. Currently, lignin from Brazilian companies is used in applications such as thermoplastics, antioxidants, phenolic resins, and dispersants [94,95,100]. In addition to being used as raw material for value-added products, these lignin-based products are a suitable substitute for the traditional fossil-based compounds. Another pilot plant for lignin processing has been operating since 2019 [101].

Lignin can also be used to produce bioplastics and polyurethanes [102]. Currently, about 50 million tons of kraft lignin are produced worldwide each year [103], but it is estimated that only 1–2% is recovered and used as raw material for products [104]. Some companies are taking advantage of the availability of this feedstock to produce bioplastics for several uses. In agriculture, for example, the use of single-use plastics in mulch films and containers for seedlings is the standard practice. These plastics cannot be recycled, and end up in landfills after one crop season. Biodegradable plastics made from lignin sourced from the wood industry are being produced to reduce the plastic pollution in the field. The advantage of lignin-based plastics over other bioplastics (e.g., from corn or potato starch) is that it takes longer to biodegrade [105], being suitable for use in agriculture. These new bioplastics from forest-based lignin are suitable for both injection molding (to produce hard plastic containers), and blown film and cast film extrusion lines, (to produce flexible packaging) [106–108].

Lignin can also be applied in the manufacture of carbon composites [109], such as carbon fibers, which can be applied in the manufacturing of different valuable products such as automobiles, aircrafts, wind turbines, among others [110]. Although carbon fibers present excellent material properties for the manufacturing of several products, the fiber's high price constrains its use. Lignin is demonstrated to be an alternative for the fossil-based polymer traditionally used as feedstock for the carbon fiber manufacturer [109]. Currently, the lignin-based carbon fiber does not reach some of the required mechanical properties to be applied in products such as aircrafts; however, it can be applied to other market segments, such as automotive and electronics manufacturers [106]. Besides significantly decreasing the fiber price, the use of lignin also offers a mitigation factor, while being suitable for application in different sectors, such as the automotive industry [111]. Carbon fiber has been appointed as one of the most profitable lignin-based products from biorefineries and, considering the increasing global demand for the products, it seems a promising market [111].

Lignin can also be used in the manufacture of hard carbon anode materials for lithium-ion batteries [112]. Lithium-ion batteries are currently extensively used in different segments, such as high-end electronics (e.g., mobile phones and laptops) and vehicles. However, challenges regarding cost effectiveness and environmental sustainability are imposed when manufacturing this product with its traditional fossil feedstock. Recent research showed opportunities to produce batteries from cellulose and lignin, which could improve the price, weight, sustainability of production and recyclable capacity of the energy storage products on the market [112]. Following this finding and market opportunities, a Finnish forest company established a pilot plant that produces bio-based carbon materials for batteries, using lignin as feedstock [113].

Chemical Derivatives from Tall Oil

Regarding the processes using wood from coniferous species, tall oil, which is one of the by-products of the kraft pulping process, can be refined into valuable chemicals, such as fatty acids, rosins and sterols [114]. In the past, the Brazilian production of crude tall oil was almost entirely exported, although there are chemical industries in the country that already include this component in their production line [115]. Currently, the industry is producing different commercial varieties of crude tall oil and even refining it into sterol compounds for the international market [115,116]. Several other chemical compounds can be produced from tall oil. One of these derivatives is bio-naphtha, which can be used in the production of biodiesel and bioplastics [117,118]. Producing chemical compounds from tall oil has many advantages, including the use of a by-product from the kraft pulping process. However, the fact that tall oil is already feedstock for many bio-based products can be considered a constraint in the development of this segment [117].

3.2.2. Solid Wood Products and Panels Industry

The Brazilian planted tree industry is one of the largest wood panel producers in the world [86], having as main products medium-density fiberboard (MDF) and high-density fiberboard (HDF), followed by particleboard and plywood [86]. Hardboard and oriented strand board (OSB) are also produced, although on a smaller scale [86]. MDF and particleboard are mostly used in furniture, while plywood and OSB are mainly used in the construction sector in floors, walls, roofs, and others [119,120]. Around 70% of the Brazilian production is directed to the domestic market [86]. In 2020, the wood panels domestic consumption increased by 3.9% from the previous year, which was attributed to the increase in home renovations related to the COVID-19 pandemic [37].

"Engineered wood" is a term used for a group of composite products that consist of wood particles, veneers or boards bound together by adhesives or other methods to form panels, beams, and planks. Different techniques are performed resulting in a wide variety of engineered wood products. Some products are relatively novel and are increasingly

being adopted by the construction sector. Examples of engineered wood products used as structural elements in buildings are: glued laminated timber (glulam), cross-laminated timber (CLT), and laminated veneer lumber (LVL). The number of companies that produce engineered wood products in Brazil has been increasing, albeit slowly, and there is a closer connection between the civil engineering and forest sectors, resulting in more wood constructions [121]. Currently, Brazil presents a varied portfolio of wood products, with emerging segments such as engineered wood products (Figure 3).

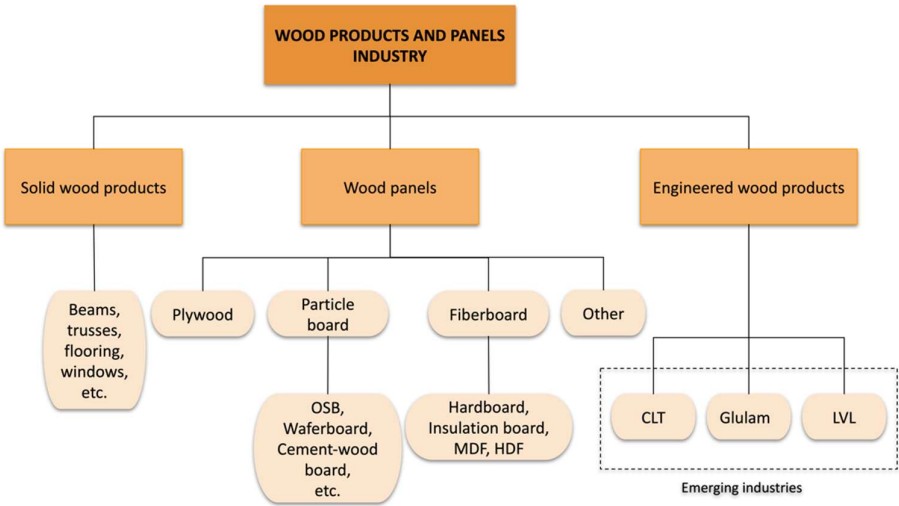

**Figure 3.** Main current products and emerging technologies from the Brazilian solid wood products and panels industry.

Worldwide, the engineered wood market is being stimulated by the construction sector, which has been integrating wood products into several construction modalities ranging from small to large scale projects, such as multi-story buildings, schools, and others [10]. Wood buildings are known for providing a comfortable indoor atmosphere and also providing efficient sound and thermal insulation [122].

Besides the aforementioned benefits, wood constructions have an important role in the circular and sustainable bioeconomy [123]. Climate change mitigation actions can be achieved when storing carbon by using wood in construction [124]. The use of sustainably-sourced wood products as a substitute to other more energy- and resources-intensive materials, that cause higher GHG emissions during manufacture, such as steel and concrete, can contribute positively to climate change mitigation [124].

Engineered wood products have many benefits when compared to non-renewable materials. However, as with any material, they are responsible for some environmental impacts. The largest climate change impacts in the production stage are due to the production of lumber [125] and energy consumption during wood drying, especially when fossil fuels are used [126–128]. However, the use of engineered wood elements for long periods can offset the GHG emissions due to carbon storage. The impact on land use and land use change from the manufacturing of engineered wood products is mostly associated with the production of roundwood and the preparation and use of roads for log transportation [125,126]. However, the production of roundwood also contributes to reducing environmental impacts on soil and groundwater as forests perform their functions of protecting the ground from soil erosion and retaining water [126]. Even considering the potential environmental impacts, using engineered wood products from sustainably-managed forests has clear advantages over mineral-based materials, such as carbon sequestration and storage, and using renewable raw material [124,129]. Some of the

engineered wood products that have been increasing in market size fostered by the wood construction sector are: glulam, CLT and LVL.

Glued Laminated Timber

Glulam, which is used as a construction material in beams and pillars, and is widely known in Europe [130], is still mostly unknown by the construction industry in Brazil. It was first introduced in the country in the 1930s, but after the initial interest by the industry, it was almost completely pushed out of the market due to the lack of development of construction standards and marketing strategies [131]. Even nowadays, its production and use are still modest compared to the European context [132]. The reason for the slow adoption of this product by the construction sector in the country is the high production costs associated with the short supply [131].

The supply issue may change in the near future. The current production capacity of the three largest companies operating in the country is around 3.5 thousand cubic meters [132], but other companies are set to start producing glulam, which will likely increase tenfold the production capacity.

Cross-Laminated Timber

The wood building market has been increasing for the past few years, and particularly the CLT segment suggests an increase in global production in the next few decades, as this technology is allowing to expand the market for timber constructions with multistory structures [133]. The global production of CLT increased from 920 thousand cubic meters in 2019 to 2 million cubic meters in 2020, being forecasted to 4 million cubic meters by 2025 with a global market value of USD 1.6 billion [130,134,135].

Brazil began producing CLT in 2012, mostly for use in single-family houses and low-rise commercial buildings [136]. The technology is compatible with multi-story buildings, where the composite product is used as structural material, such as beams and pillars [121]. Currently, a new plant, with production capacity of 60 thousand cubic meters, is being built in Brazil, which will produce CLT and glulam using pine wood [137].

Laminated Veneer Lumber

LVL has been the focus of research studies in Brazil since the 1990s, although it was only in 2016 that it started to be commercially produced in the country [121,138]. This composite material is frequently used in floors, walls, rafters, beams, among others. The interest in LVL has been increasing due to its popularization as a construction material in high rise buildings. The European market represents a major market, and it is projected to continue a 6% average annual growth in consumption [139].

Although LVL has been produced in Brazil, its applications in building construction are increasing modestly [140]. However, the installed capacity of LVL production in Brazil is around 100 thousand cubic meters per year [121]. Even though LVL elements present satisfactory structural performance [141], it is not commonly used because consumers are unfamiliar with the product [142]. However, it is possible that LVL will gain popularity in Brazil due to its versatility as a material and the superior physical properties compared to solid wood [141]. Moreover, it can be produced with either coniferous or non-coniferous species, such as pine, eucalypt, or the many indigenous species [140,143].

3.2.3. Biorefineries

Besides the previously mentioned biochemicals produced by the pulp and paper sector, there are many chemical compounds that can be obtained from non-wood forest products, such as resins and tannins (Figure 4). These can be used in several industries, as ingredients for food, beauty products, and pharmaceuticals, among others.

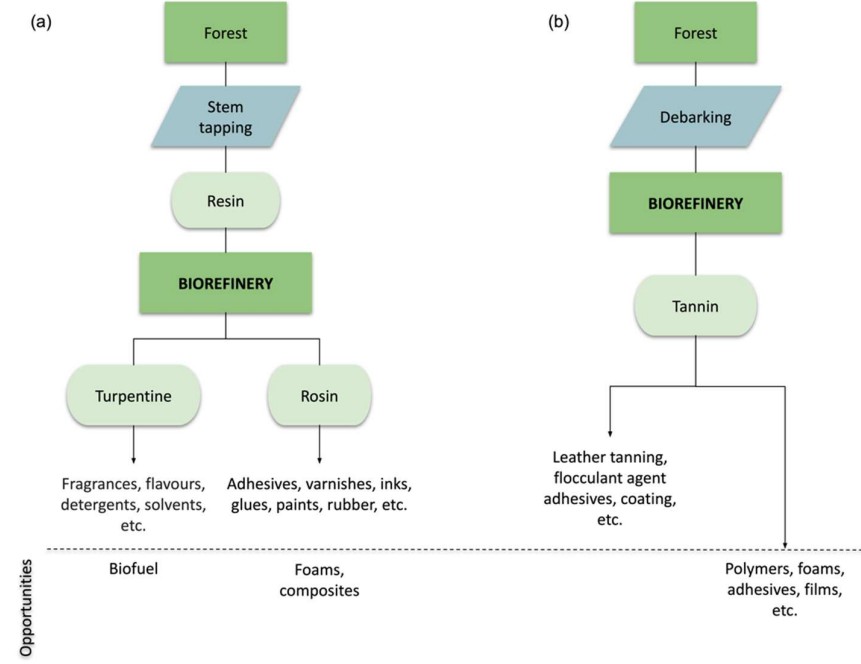

**Figure 4.** Simplified value chains for (**a**) tree resin and (**b**) tannins from the Brazilian planted tree industry.

Even though resin and tannins are manufactured into several products, companies may seize future opportunities by expanding their product portfolio and including higher value-added products, such as diverse bio-chemicals and bio-based polymers.

Resin Derivatives

The global resin market is highly concentrated, having China, Brazil, and Indonesia covering over 90% of the world's production, with Brazil having the highest extraction rate: 3.8 kilograms per year from a single tree, on average or 4.2 tons per hectare annually [144]. Between 2017 and2018, Brazil produced 186 thousand tons of resin [145]. Due to forest resource and labor availability, combined with a high production yield, Brazil still has the potential to increase production[144]. In 2018, Brazil exported nearly 27 thousand tons of pine resin, and about 70% of this volume was exported to Portugal, 12% to Vietnam, 9% to China, and 9% to several other countries [146].

The chemical derivatives from resin are used in the manufacture of hundreds of products in the chemical and food industries as ingredients for disinfectants, detergents, paints, adhesives, flavorings, among others [147,148]. Even though there are many tree species that produce resin, most of the commercial natural resin comes from pine trees. The two most common chemicals obtained from pine resin are turpentine and rosin [148]. These intermediate products can be further broken down into several chemical compounds that are used in many industries to produce paints, fragrances, cleaning products, inks and paints, adhesives, pharmaceuticals, among many other applications [149].

The wide potential for chemical modification characterizes rosin as a valuable material for the development of diverse applications [150]. Research indicates that rosin can be a feedstock to materials such as insulation foam and structural plastics [151]. Certain rosin-based materials also present pharmaceutical properties, where applications as film-coating and encapsulation of drugs have been studied [152]. Turpentine is also a valuable material used in diverse industries, especially in fragrance applications as this component presents unique chemical derivatives for the perfumes, cosmetics, and food industry. Considering a shift from fossil-based to bio-sourced raw materials used in the mentioned industries, terpenes are predicted to gain importance and are expected to be one of the most

strategic materials for these industries [153]. Biochemicals such as rosin and turpentine derivatives can be transformed into a large array of chemicals that are used in diverse industries and that can attract market interest. For this reason, the growth of the global pine chemical market has been observed in the past few years [154].

Tannin Derivatives

The Brazilian tannins sector is concentrated in the southern region, where most processing companies are located. In 2019, around 41 thousand tons of tannins were produced, of which nearly 25 thousand tons were exported to India (26%), Mexico (15%), China (13%) and several others [146,155]. Special variations in tannins are currently used in several industries (e.g., food, panel, and clothing), in water treatment, and for other applications [156]. The feedstock used to manufacture tannins is the bark of black wattle (*Acacia mearnsii* De Wild.). In 2019, there were 76 thousand hectares of black wattle planted in southern Brazil, where it is mainly cultivated [155].

Besides the aforementioned uses, tannins have been proven to be a valuable biosource for the production of several higher value-added products, such as polymeric composites, foams, films, among others [157]. Tannins from black wattle were successfully used as a compatibilizer agent in wood-plastic composites, which are used for construction compartments, packaging, automotive components, among others [158]. Tannins can also be used in films made in a nanofibrillated cellulose matrix, where they can provide chemical protection. This type of film presents good barrier properties, such as hydrophobicity, antioxidant activity, and resistance against solvents. The nanofibrillated cellulose films with tannin can be used in the manufacture of products known as "active packaging", used for packaging of food and pharmaceuticals [159]. Further bioactive properties were identified in the application of tannins as a coating agent, where they exhibit an antimicrobial effect on hospital supplies [160].

Tannins are also applied to thermosetting structural foams, used worldwide in packaging, building materials, automobiles, cushioning, insulation, marine structures, and electronic applications [161]. In this use, foams are obtained through tannin's ability to perform polyaddition reactions with furfuryl alcohol for polymerization in acid catalysis [162]. For use in these applications, the foams must necessarily have some resistance to fire, generating as little smoke and toxic products as possible. In comparison with polyurethane, polyvinyl, and polystyrene foams, tannin foams perform better in terms of fire resistance. Future research is being directed towards the replacement of formaldehyde in foams, in order to reduce the reliance on fossil sources [161,162].

Derivatives from Cellulosic Sugar

Currently, biochemicals are used in several sectors, such as construction, textiles, food, animal feed, pharmaceuticals, and cosmetics. Polymers, such as polyethylene glycol, are industrially relevant products that can be manufactured from biomass [163]. Polyethylene, a polymer used for plastics, is usually produced from fossil sources. Bio-based polyethylene can be used in the production of bioplastics that have the same properties as conventional fossil-based plastics. The bioplastic sector shows significant economic potential, as it is predicted to grow rapidly over the next few years, reaching 7.6 million tons by 2026, around three times the production volume of 2021 [164].

Biorefineries can manufacture a diverse portfolio of products from cellulosic biomass, used in several industry sectors (Figure 5).

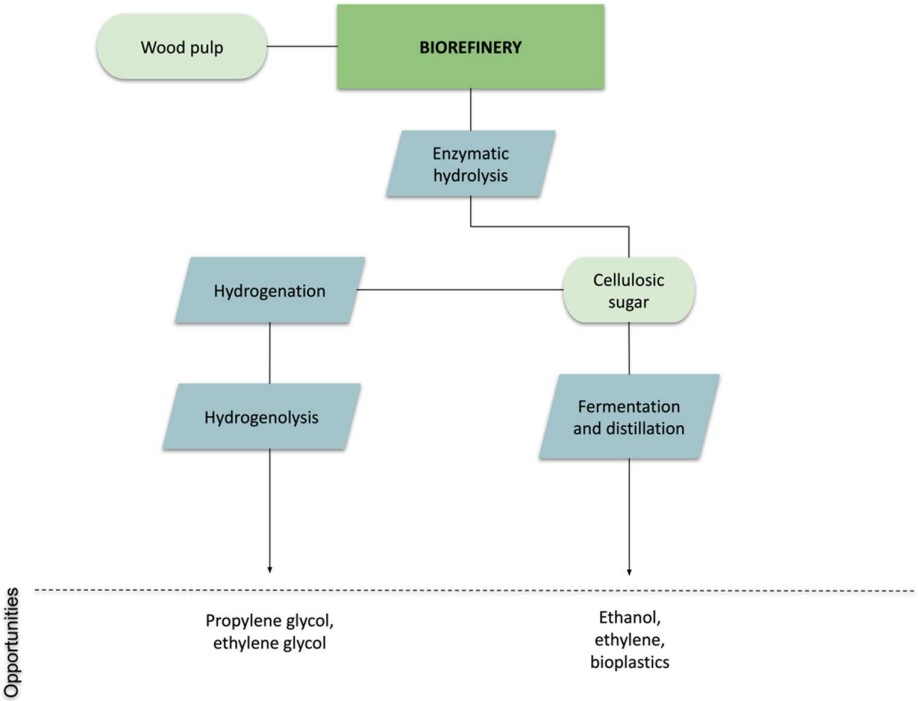

**Figure 5.** Simplified production process of cellulosic sugar and potential products.

Brazil currently produces bio-based polyethylene from sugarcane biomass [165,166]. The country has also started manufacturing glycol, another important polymer, in a pilot plant using a sugar-based production process [167]. This polymer is one of the most important components in the chemical industry and is traditionally derived from petroleum. Glycols, more specifically monoethylene and monopropylene glycols, can also be efficiently produced from lignocellulosic biomass, such as wood [168]. These chemical compounds can be used as raw material in several types of products, including resins and detergents, as well as polyester-based products such as bottles, packaging, and textiles [169].

Liquid Biofuels

Liquid biofuels are a technically feasible alternative to the utilization of non-renewable fuels used in the transportation sector, especially for aviation, ships, and trucks [170]. Biofuels are seen as carbon neutral as the biomass carbon comes from the atmosphere, being released when burned, to be sequestered again by the growing biomass stock [171]. However, the assessment of sustainability of biofuels is very complex due to the variability in inputs and outputs since feedstock production and processing happen in different global regions [172].

Brazil has a large market developed for biofuels; however, this market is based on agricultural crops, especially sugar cane [51,173]. Ethanol from sugar cane is the main product of the Brazilian bioeconomy [174]. Since the 1970s, the Brazilian government has established policies to incentivize the production and consumption of ethanol instead of fossil fuels [174]. The increase in the use of ethanol as fuel could be observed throughout the years, with 17 million cubic meters of hydrated alcohol being consumed in 2010 and 24 million cubic meters in 2019 [175]. In 2019, 49% of the total volume of fuels used by light vehicles was represented by ethanol [176].

Even though ethanol production is currently based on agricultural feedstock, the production based on forest biomass, such as eucalypt, has proven to be technically feasible and efficient [177–180]. According to estimates, one ton of eucalypt bark can yield nearly

200 liters of ethanol [180]. One of the greatest advantages of using residues from the forest industry is that there is no competition for feedstock with cellulosic pulp, food, and feed production [179,180]. Besides using ethanol as fuel for light vehicles, sustainable aviation fuels can also be successfully manufactured from forest biomass. It is estimated that around 26% of the national demand for sustainable aviation fuels can be obtained from the processing of wood residues from eucalypt plantations in biorefineries [181].

## 4. Discussion

Forests can offer many benefits; however, the sustainable provision of forest products depends on the type of forest and its management system [22,182]. Planted forests that are sustainably managed provide raw materials for the manufacturing of products that can potentially contribute to climate change mitigation. This benefit can be achieved by long-term carbon storage in wood products (e.g., wood buildings and furniture) or by the substitution for non-renewable and GHG emissions-intensive products (e.g., bioplastics replacing fossil-based plastics).

Like many other countries, Brazil's bioeconomy strategy aims to promote sustainable development. The definition of bioeconomy adopted in the Brazilian strategy seems to focus on biotechnological processes and innovations to substitute the use of fossil-based materials [40]. However, it does not focus on any specific biomass source, which could limit the development of new forest products due to the use of well-established technologies for agricultural feedstock.

Brazil currently focuses on the export of commodities, such as low-value-added products with large demands [27,183]. Despite the importance and strength of its agribusiness sector, Brazil is ranked only as the 21st exporter economy in the world [184]. There is potential for the forest bioeconomy to grow both in quantity and quality of products and services, but it is highlighted especially in terms of value added, with innovation in biosciences needed to obtain a more competitive advantage [51,183]. The forest biomass value pyramid indicated that new products from the Brazilian forest bioeconomy are aiming at high-value-added products. However, as the country still relies on commodity markets, this hinders investments and implementation of actions in the structural and institutional spheres, making it harder to support innovation and promote an economy based on high-value-added products [27].

In recent years, the global pulp and paper market has been experiencing major changes, mainly associated with the decline in demand for newsprint, printing, and writing paper [185,186]. In line with this global trend, the production of writing and printing paper has also been declining in Brazil since 2012, although the production of wood pulp has been increasing [86]. In 2020, Brazil produced 21.5 million tons of wood pulp, of which 76% (or 16.5 million tons) was exported to Asia, Europe, and other regions[86]. Following the notion that Brazil should reduce the exports of low-value-added products to boost its bioeconomy, the country could use part of this potential feedstock volume for high value-added products.

Pulp mills can be integrated with other industries, such as biorefineries, which can produce a range of high value-added biomaterials, biofuels, and biochemicals. Brazil has the opportunity to further the development of biotechnology, especially if focused on industrial processes and products [187]. This would be in line with Brazil's National Bioeconomy Strategy [52], which has as one of the main goals to enhance the industrial aspect of Brazil's bioeconomy sector. Nowadays, companies mainly gain competitive advantages and economic benefits from product innovation, product portfolio diversification, or by focusing on adding value to products already produced or manufactured [23,188,189]. Adding value can be achieved by improving the main production systems (e.g., producing NCC from wood pulp) or by using by-products of industrial processes as feedstock for high-value-added materials (e.g., lignin-based products). While improvements in the main production systems could benefit the fiber sectors (such as paper, packaging, and

textile industries), using by-products as feedstock could also benefit biochemicals and bio-fuel segments, increasing the overall profitability in the value chain [190].

The Brazilian forest sector could also benefit from the increase in the degree of processing of their main products. Even though the sector has demonstrated a significant decrease in raw material exports in the past few years, the country still struggles to achieve a higher level of industrial processing, exporting mostly products with a low degree of transformation [46]. An example could be dissolving cellulose, which is currently mainly exported as an intermediate product to be used as an input and processed into a down-stream value-added product in another country [86]. Most of the value of a product tends to be added on the further processing of the raw material [38,191]. Therefore, investing in an advanced processing industry could enhance the forest bioeconomy in Brazil. The country's pulp and paper industry is working towards the production of NCC, which is being highlighted as a material useful for several industry segments [55]. This effort seems to be aligned with international market trends, as the demand for NCC is forecasted to increase in the following years [192].

The development of engineered wood products in the country is still in the early stages. However, the global market for engineered wood is expanding [130,134], which could not only be an opportunity for the domestic market, but also for the international market. Brazil's export of low processed wood products, such as roundwood and sawn wood, has been increasing. In the period of 2010 to 2020, the export volume of these products increased from 1.4 to 3.5 million cubic meters[86]. If the country reduced the export volumes of low value-added products such as roundwood and sawn wood, at least part of 3.5 million cubic meters would be available for further processing into engineered wood products, promoting a higher added-value attribution when compared to the current products. Besides changes in the market, the expansion of the engineered wood products industry in Brazil could possibly provide positive impacts on environmental, health and social aspects related to the development of wood building systems [193].

The Brazilian forest sector produces and processes high-quality biomass; however, it has shown a slow adoption of systems with integrated biorefineries [194,195]. Instead of focusing on producing value-added biomaterials, such as biochemicals and biofuels, the focus is still on selling the energy resulting from the burning of the by-products and residues from the cellulosic pulp production [196]. Biorefineries supplied with forest biomass could be an avenue for the development of the Brazilian bioeconomy, since biomass from planted forests can be supplied throughout the year and is not much affected by the growing season, as is the case for biomass from agriculture [197]. In addition, commercial forests can be planted in degraded lands, which means less competition for lands that can be used for food production, as normally agriculture plantations require lands of higher quality than forest plantations [197].

Regarding the biorefineries that process non-wood forest products, such as tannins and resin, the activities in this industry could be expanded following the growth in interest and the market [154,198], contributing to the development of the forest-based bioeconomy in the country. More than half of the tannin produced in Brazil is exported [146], which could potentially be used in many industries (e.g., preservatives and additives in the food industry) and is usually turned into products of high value and low volume [199]. Considering that South America is an important supplier for European and North American markets [200], the production of further processed products could represent a promising future for the segment. The same can be said about valuable biochemicals derived from resin, as biochemicals are a growing market [198].

Based on the high productivity of planted forests in Brazil and the established market in the country, biofuels and other biochemicals could be considered important for the development of the forest-based bioeconomy. Even though biorefineries are currently focusing on using annual crops, the forest industry can offer many feedstock options, especially by better using undervalued by-products, residues, and waste. Crude sulfate turpentine, which is currently used for fragrances and solvents, among others, is being considered as

a technically viable raw material for biodiesel production [201,202]. Efforts towards the innovative application of by-products in the Brazilian forest industry are related to lignin processing. Many products are already manufactured from lignin, most of which substitute fossil-based products, representing a good opportunity in both the market and environmental aspects [201].

Based on the identification of the current efforts already happening in the country, the feedstock availability, and the country's bioeconomy action plan, it is possible to highlight the abovementioned sectors as the most prosperous sectors for the development of the forest bioeconomy in Brazil. Overall, market data (e.g., production, demand) on the reviewed products were scattered and usually unavailable, especially when targeting only Brazil. Most of the trustworthy commonly known databases report only traditional forest products, which could potentially hinder the analysis of the new products from a forest-based bioeconomy. Therefore, making data on the innovative forest products market available would be crucial to allow for a more in-depth assessment of the development of the country's bioeconomy.

## 5. Conclusions

Brazil is currently one of the largest forest biomass producers, playing an important role in the global forest products markets. The pulp and paper industry, which is the main consumer of raw materials from planted forests in the country, has been showing interest in processing its by-products (e.g., lignin and tall oil) in addition to further developing other products, such as dissolving pulp and NCC. In addition to products from pulp and paper side streams, biorefineries are increasingly processing non-woody biomass, such as tannins and resin, into value-added chemicals. Other derivatives from cellulosic sugar, such as ethanol and bioplastics, are technically feasible but not yet produced from forest feedstock in Brazil. The solid wood products and panels industry has shown some developments associated with the production of engineered wood products, which are being highlighted as an alternative to the use of non-renewable construction materials, such as bricks, cement, and steel.

Our study covered many of the recent developments from the most relevant industries of the Brazilian planted forest sector and shed some light on the possible avenues for these industries, in terms of innovative products. However, there are still many challenges and opportunities for the development of the forest bioeconomy in the country, which could enhance the sustainability aspects of the forest sector and the country's competitiveness.

**Author Contributions:** Conceptualization, Y.I.M. and M.H.; methodology, Y.I.M. and M.H.; investigation, Y.I.M. and M.H.; writing—original draft preparation Y.I.M, M.H. and P.J.V.; writing—review and editing Y.I.M, M.H, P.J.V. and A.L.M. All authors have read and agreed to the published version of the manuscript.

**Funding:** This research received no external funding.

**Institutional Review Board Statement:** Not applicable.

**Informed Consent Statement:** Not applicable.

**Data Availability Statement:** Not applicable.

**Conflicts of Interest:** The authors declare no conflict of interest.

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
