# Peer review of "Forest Bioeconomy in Brazil: Potential Innovative Products from the Forest Sector"

_land, doi:10.3390/land11081297_

Round 1

Reviewer 1 Report

The manuscript provides a broad and complete overview of the bioeconomy of the Brazilian forest sector. The quality of this manuscript, valuable in itself, can be increased if 2 more aspects are highlighted.

1. how is the forestry sector under consideration distributed across Brazil? Are there regional focuses? If so, what is the particular strength of these focal points?

2. different products that could enable the further development of the forest industry are discussed. What are the requirements to drive this further development?

Also: in line 32, 591, definitions of the bioeconomy are cited that limit the bioeconomy to the application of biotechnological methods. Does this also apply to the Brazilian bioeconomy strategy and if so, why? Why are chemical and physical methods excluded?

Line 418, 432: Please do not use abbreviations in headlines.

Line 626; 643: Where are the hurdles to get into higher processing levels?

Reviewer 2 Report

The paper presents a good and extensive overview of possible process and application pathways for feedstock originating in Brazilian forests. The paper also to some extent analyses the market potential for resulting bio-based materials and bio-based products. However, the paper fully misses any assessment of potential environmental and socioeconomic impacts to land (and soil) if those pathways are followed. More generally, a lifecycle thinking approach is missing in the paper setting boundaries to the upscaling of the identified pathways. 

In conclusion, in my view the article could not be published in Land in its current form. A major revision with a lifecycle thinking approach to all identified pathways would be needed for reconsideration.

Reviewer 3 Report

Dear Authors,

I have read your paper with great interest.

It represent a valuable effort to advance the current Brazilian planted forest bioeconomy to a sustainable and circular version.

Yet, the research is missing its quantifiable results. While the options for novel bio-based products are clustered, one can not judge what are the most prosperous sectors, what markets to focus on, if the cascading use of biomass is in place, what production will shrink if this novel value chain is established, what are the high/low value; bulk or niche bio-based products etc. Section 3.3.2.2. gives a bit of the idea but cannot be compared with the alternatives. Maybe to organize the novel and innovative products by market potential or along the bioeconomy product pyramid? Not sure how to address this issue but that would be my main remark and the reason to ask for major revision. Your paper has to make a step forward to bring novel information to the scientific community, which it fails to do in the current state.

However, there are some general remarks that have to be considered before re-submitting the paper:

1. Please, make sure to use passive voice instead of "we".

2. There are many sections that could be shortened and supported by figures, values or credentials. The paper is sometimes written in grey literature style using words such as "growth at an astonishing pace" (L422), L419-21 has little scientific value, etc.

3. Along the paper, it is not clear if the figures are related to the planted forests industry sector or in general (e.g. pulp and wood, MDF etc.). Please make sure it is always a figure reflecting planted forest industry sector in Brazil.

4. I did my best to suggest alterations in the text to help you direct the revision. Please, consider the insertion as idea and not as a finite intervention.

Good luck!

Round 2

Reviewer 2 Report

Dear authors,

Thanks for following my suggestions.

Author Response

Thank you again for the time taken to review our work. 

Reviewer 3 Report

Dear authors,

thank you for your efforts, the revised paper is much advanced than the original. 

I still find the Results and Conclusions part lacking new knowledge, except for inventorizing and clustering possible novel wood-based products. The clustering seems to be the most important part of your paper but it could be applied to any other country. By reading your paper, the reader is not gaining insight what the most lucrative / sustainable / promising / attractive novel forest products would be for Brazil. 

There are many ways how to give the insight and you are not mandated to follow my lead but how about that you build your Results upon Figure 1? How about positioning all the identified novel products in the pyramid or by each cluster, with giving the quantities of biomass available on the right side of the pyramid? That could give some idea of relative value of those novel products to the reader as well as corresponding quantities for each potential product.  

I apologize, but I can give you a recommendation before Results and Discussion sections are strengthened for new knowledge. However, when reviewing the original, I was between "reject" and "major revision". I acknowledge your effort to improve the material. Now I am confident that you can get this paper published. 

Best regards and good luck.
